# The Landscape of Cytogenetic Aberrations in Light-Chain Amyloidosis with or without Coexistent Multiple Myeloma

**DOI:** 10.3390/jcm12041624

**Published:** 2023-02-17

**Authors:** Haiyan He, Jing Lu, Wanting Qiang, Jin Liu, Aibin Liang, Juan Du

**Affiliations:** 1Shanghai Tongji Hospital, School of Medicine, Tongji University, Shanghai 200092, China; 2Department of Hematology, Myeloma & Lymphoma Center, Changzheng Hospital, Naval Medical University, Shanghai 200003, China

**Keywords:** amyloidosis, FISH, prognosis

## Abstract

Interphase fluorescence in situ hybridization (iFISH) has been well established in the preliminary prognostic evaluation of multiple myeloma (MM). However, the chromosomal aberrations in patients with systemic light-chain amyloidosis, notably in patients with coexistent MM, have been rarely investigated. This study aimed to evaluate the effect of iFISH aberrations on the prognosis of systemic light-chain amyloidosis (AL) with and without concurrent MM. The iFISH results and clinical characteristics of 142 patients with systemic light-chain amyloidosis were analyzed, and survival analysis was conducted. Among the 142 patients, 80 patients had AL amyloidosis alone, and the other 62 patients had concurrent MM. The incidence rate of 13q deletion, t(4;14), was higher in AL amyloidosis patients with concurrent MM than that of primary AL amyloidosis patients (27.4% vs. 12.5%, and 12.9% vs. 5.0%, respectively), and the incidence rate of t(11;14) in primary AL amyloidosis patients was higher than that in AL amyloidosis patients with concurrent MM (15.0% vs. 9.7%). Moreover, the two groups had the similar incidence rates of 1q21 gain (53.8% and 56.5%, respectively). The result of the survival analysis suggested that patients with t(11;14) and 1q21 gain had a shorter median overall survival (OS) and progression-free survival (PFS), irrespective of the presence or absence of MM, and patients with AL amyloidosis and concurrent MM carrying t(11;14) had the poorest prognosis, with a median OS time of 8.1 months.

## 1. Introduction

Immunoglobulin light-chain amyloidosis (AL amyloidosis) is a rare clonal-plasma-cell disorder, that leads to organ damage by circulating light chains and their deposition as amyloids [1]. When the above AL amyloidosis patients also have myeloma-defining events, including hyper-calcemia, renal failure, anemia and lytic bone lesions (CRAB criteria), caused by clonal proliferation of plasma cells, they are considered to have concurrent multiple myeloma (MM) [2]. Some chromosomal abnormalities (CAs) in AL amyloidosis and MM patients, using iFISH, have been described (e.g., t(11;14), t(4;14), 1q21gain, -13/13q deletion, and 17p deletion) [3,4,5]. However, the frequencies of the respective chromosomal aberrations are controversial. AL amyloidosis patients have been found with a higher frequency of t(11;14) and less frequent 1q21 gain compared to MM. The t(4;14) and 17p deletion are rare in AL amyloidosis patients, and the incidence rates of -13/13q deletion are similar in AL amyloidosis and MM [3,4,5]. In MM, t(11;14) and -13/13q deletion were found as standard prognostic factors, and t(4;14), 1q21 gain, and 17p deletion were established as adverse prognostic factors [6,7,8]. Alongside that, t(11;14) in AL amyloidosis has been found to be correlated with a poor prognosis [9,10]. There have been limited studies on the association of FISH results and prognosis of AL amyloidosis, and rare studies have found the prognostic significance of FISH results in AL amyloidosis patients with concurrent MM. This study aimed to identify the most relevant FISH biomarkers present in patients with AL amyloidosis alone or with concurrent MM.

## 2. Materials and Methods

### 2.1. Patients and Treatment

A consecutive series of 142 newly diagnosed AL amyloidosis patients, who presented to our hospital from November 2012 to December 2021, were analyzed retrospectively. A total of 80 of the 142 newly diagnosed AL amyloidosis patients had primary AL amyloidosis, and the other 62 patients were AL amyloidosis patients with concurrent MM. The diagnosis of AL amyloidosis was confirmed by a Congo-red-positive biopsy and immunohistochemistry of the amyloid [1]. Meanwhile, the presence of the kappa or lambda chain was confirmed through immunohistochemistry and/or immunofluorescence. Moreover, monoclonal plasma cell proliferation was investigated by multi-color flow cytometry on aspirated bone marrow cells. Patients were evaluated for the presence of symptomatic MM based on the International Myeloma Working Group (IMWG) criteria [11]: bone marrow clonal plasma cells > 10% and evidence of myeloma-defining events, including the slim-CRAB criteria. Patients with AL amyloidosis with >10% bone marrow plasma cells and any myeloma-defining events were described with coexistent MM. iFISH was performed for all the patients as part of their routine clinical testing. Patients gave written informed consent for the iFISH and data analysis in accordance with the Declaration of Helsinki. Approval was obtained by the Ethics Committee of Shanghai Changzheng Hospital.

Among the 80 primary AL amyloidosis patients, 71 patients were treated with bortezomib-based regimens (cyclophosphamide/bortezomib/dexmethasome (CBd) or bortezomib/dexamethasone (Vd)), five patients treated with lenalidomide-based regimens (lenalidomide/cyclophosphamide/dexmethasome (RCd) or lenalidomide/dexmethasome (Rd)), three patients were treated with thalidomide-based regimens (thalidomide/ cyclophosphamide/dexmethasome (TCd)), and one patient was treated with ixazomib-based regimens (ixazomib/dexmethasome (Id)) for induction therapy, and three of the 80 patients received autologous transplants as a consolidation therapy. Among the 62 AL amyloidosis patients with concurrent MM, 47 patients were treated with bortezomib-based regimens (CBd or Vd), seven patients were treated with lenalidomide-based regimens (RCd or Rd), five patients were treated with bortezomib combined with lenalidomide (bortezomib/lenalidomide/dexmethasome (VRd)), and three patients were treated with daratumumab-based regimens (daratumumab/bortezomib/dexmethasome (DVd)). A total of six of the 62 patients received autologous transplants as a consolidation therapy. None of the 142 patients received venetoclax.

### 2.2. Interphase Fluorescence In Situ Hybridization

iFISH was performed on CD138-positive bone marrow (BM) plasma cells, purified by auto-magnetic-activated cell sorting with anti-CD138 immunobeads, according to the description of previous studies [12]. D13S319, 17p13.1 (P53), and 1q21 (CKS1B) probes were used to examine 13q deletion, 17p deletion, and 1q21 gain, respectively. An IGH fusion probe was adopted to detect the translocations involving IGH. Moreover, LSI (locus-specific identifier) IGH/CCND1, LSI *IGH/FGFR3*, and LSI IGH/MAF were employed to detect the partners of IGH, including: t(11;14) (q13;q32), t(4;14) (p16;q32), and t(14;16) (q32;q23). The cut-off values for the respective probe in iFISH included 10% for fusion and 20% for numerical abnormalities, in accordance with the European Myeloma Network (EMN) FISH workshops recommendations [13].

### 2.3. Statistical Analysis

OS was defined as the time from diagnosis to the last contact or death. PFS was defined as the time from the initiation of treatment to disease progression, relapse, and death, for any cause. R (version 4.1.2) was adopted to perform all analyses. The comparison among groups and the comparison between two groups were drawn using the one-way ANOVA, through the chi-square test. Survival curves were plotted using the Kaplan–Meier method, and survival was compared through the log-rank test. Univariate and multivariate analyses were conducted using Cox regression. Factors with *p* values of <0.05 in univariate analysis were included in the multivariate analysis. A *p* value of <0.05 indicated a difference achieving statistical significance.

## 3. Results

### 3.1. Baseline Clinical Characteristics

The baseline characteristics of each group are listed in Table 1. A total of 80 patients had primary AL amyloidosis, and the other 62 patients were AL amyloidosis patients with concurrent MM. No significant differences between the two groups were found in the age and gender ratio. The median age was 62 years for both groups, and the proportions of male were 70.0% and 61.3%, respectively. A significant difference was found in the proportion of the M protein type. IgG-type disease was significantly higher in AL amyloidosis with concurrent MM, while the light-chain type was higher in the group of AL amyloidosis alone. There were four cases of IgD type disease in AL amyloidosis with concurrent MM, but none in primary AL amyloidosis. A higher proportion of patients with AL amyloidosis and concurrent MM were classified into II or III stage, in accordance with the Mayo 2012 staging system for the higher light-chain burden. Lower levels of hemoglobin and higher levels of β_2_ microglobulin were observed in AL amyloidosis with concurrent MM patients for their higher tumor burden.

### 3.2. Frequencies and Distributions of FISH Abnormalities

Of the 142 patients, 121 (85.2%) patients had at least one abnormality demonstrated by FISH. Table 2 lists the frequencies of each FISH abnormality in the two groups. The incidence rates of t(11;14) in AL amyloidosis patients was higher than that of AL amyloidosis patients with concurrent MM, however, no statistical significance was found. The incidence rate of t(4;14) and 13q deletion was higher in AL amyloidosis patients with concurrent MM compared with that of patients with AL amyloidosis alone, however, the only the statistical difference found was in 13q deletion. A total of 78 of the 142 patients had an increased 1q21 copy number. The incidence of 17p deletion and 1q21 gain were similar between the two groups, and t(14;16) was positive only in two patients with AL amyloidosis and concurrent MM, and none in those with AL amyloidosis alone. Figure 1 indicated the distributions of FISH abnormalities.

### 3.3. Survival and Prognosis Analysis

The median follow-up time was 21 months and 142 AL patients were included in survival analysis. The Kaplan–Meier curves of overall survival (OS) and progression-free survival (PFS) in AL amyloidosis patients with or without concurrent MM are presented in Figure 2A,B. The median OS time was not reached and the median PFS time was 67.1 months for AL amyloidosis patients with concurrent myeloma, while the median OS and PFS times for patients with AL amyloidosis alone were 39.5 months and 20.7 months, respectively. A significant difference was found between the two groups in OS and PFS, with *p* values of 0.037 and 0.021, respectively. We further explored the prognosis of each FISH abnormality, including -13/13q deletion, 1q21 gain, t(11;14) and t(4;14) in patients with AL amyloidosis alone or AL amyloidosis with concurrent MM. The prognosis of t(14;16) and 17p deletion was not included in the survival analysis for the limited sample size. We found that t(11;14) and 1q21 gain were unfavorable prognostic factors, both in patients with AL amyloidosis alone or patients with AL amyloidosis and concurrent MM. Patients with AL amyloidosis and concurrent MM carrying t(11;14) had the poorest prognosis, with a median OS time of 8.1 months, while -13/13q deletion only had unfavorable prognostic influence on PFS, but not OS, and t(4;14) had no prognostic significance on both the PFS and OS of the two groups. The Kaplan–Meier curves are shown in Figure 2C–J.

## 4. Discussion

The risk stratification and prognosis of AL amyloidosis is primarily based on the severity of organ involvement, especially cardiac involvement and FLC levels [14]. Unlike MM, that is heavily dependent on cytogenetic aberrations for risk stratification [7], there have been rarely established cytogenetic prognostic markers for AL thus far. Both entities shared the same chromosomal aberrations and similar clustering patterns, whereas rare studies have investigated the cytogenetic features of patients with AL amyloidosis and concomitant MM. In this study, the chromosomal abnormalities were investigated by iFISH analysis in AL amyloidosis patients with or without concurrent MM. The incidence rate of t(11;14) in the patients with AL amyloidosis was examined as 15.0%, lower than that reported previously (32~62%) [4,5,9,10,15,16,17,18]. Only one study reported a similar incidence rate of t(11;14), at 20% [19]. The relatively low incidence rate in this study may be due to the racial difference, therefore more reports of Asian populations should be conducted. The incidence rate of t(11;14) was obtained as 9.7% in patients with AL amyloidosis and concurrent MM, close to that of MM patients without AL amyloidosis, as reported in our previous study [12]. Muchtar et al. suggested a lower deep-response rate in patients with t(11;14) treated with both bortezomib and immunomodulatory-based regimens compared with those lacking t(11;14), thus corresponding to an inferior overall survival [10]. Additionally, the adverse effect of t(11;14) on prognosis has been confirmed in several studies [7,9,15]. In this study, most of the patients included were treated with bortezomib-based treatment. An unfavorable prognosis of t(11;14) was found in both groups, especially AL amyloidosis patients with concurrent MM carrying t(11;14), as they had the shortest survival time. Translocation (11;14) or the respective cyclin D overexpression leads to an inferior prognosis in AL amyloidosis [20]. In MM, t(11;14) was a translocation in chromosomes 11 and 14, leading to a fusion of the genes IGH and CCND1 [21]. Whether a different splicing exon or mutation in the fusion gene exists in amyloidosis patients should be further explored. Recent trials on the use of venetoclax, an oral B-cell lymphoma BCL-2 protein inhibitor, has shown specific activity in the myeloma subset of patients with the t(11;14) [22], and a trial on the use of venetoclax in amyloidosis patients was also performed [23], which highlights the therapy selection specifically based on FISH abnormalities.

The prevalence of 1q21 gain was 0–36% in patients with AL amyloidosis alone [4,9,16,19,24,25], while two studies reported the incidence rate of 1q21 gain were 50% and 67% in patients with AL amyloidosis and concurrent MM [4,17]. However, the two studies have a very limited sample size. In this study, the incidence rates of 1q21 gain were 53.8% and 56.5% in the two groups, respectively, similar to the incidence rate in MM without AL amyloidosis, and in which 1q21 gain was reported as an unfavorable prognostic factor [6,7]. Bochtler et al. observed an inferior survival in melphalan-treated AL amyloidosis patients with 1q21 gain. In their series, patients with 1q21 gain showed a myeloma phenotype, and 61% of the 1q21 gain patients had significantly higher bone marrow FLCs and bone marrow plasmacytosis, as well as higher rates of intact heavy chain [24]. Another study from Bochtler et al. found that the gain of 1q21 conferred no adverse prognosis in this bortezomib–dexamethasone-treated group [9]. A potential adverse prognostic effect of concomitant gain of 1q and deletion of 14q was found from a genome-wide copy number array analysis [25]. According to a recent study, the presence of 1q21 gain was found to be independently correlated with major organ deterioration, progression-free survival (MOD-PFS) and OS on a multivariate analysis in AL amyloidosis patients treated with daratumumab [26]. There have been limited data about the prognostic significance of 1q21 gain in AL amyloidosis, especially in the entity of AL amyloidosis with concomitant MM. In this study, most of patients were treated with PI-based regimens. An adverse effect of 1q21 gain was also found in our work, and the AL patients with concurrent MM, carrying 1q21 gain, typically had the poorest prognosis.

The prevalence of t(4;14) and 17p deletion was very low in AL amyloidosis in previous reports. There have been rare reports on their prognostic significance in AL amyloidosis for a limited number of cases. A retrospective study reported 44 AL patients with 17p deletion from seven countries, and the median overall survival for the above patients was 49 months [27]. In this study, only four AL patients, harboring 17p deletion, were found. The prevalence of t(4;14) in AL amyloidosis patients, with or without MM, was determined to be 12.9% and 5%, respectively, and t(4;14) did not have any effect on the prognosis of either group of patients. However, due to the limited sample size, its prognostic significance should be explored in larger groups.

## 5. Conclusions

In brief, the FISH results and clinical characteristics of AL amyloidosis patients, with and without MM, were analyzed. The results suggest that the presence of t(11;14) and 1q21 gain tend be correlated with a poor prognosis, especially in the AL amyloidosis patients with MM.

## Figures and Tables

**Figure 1 jcm-12-01624-f001:**
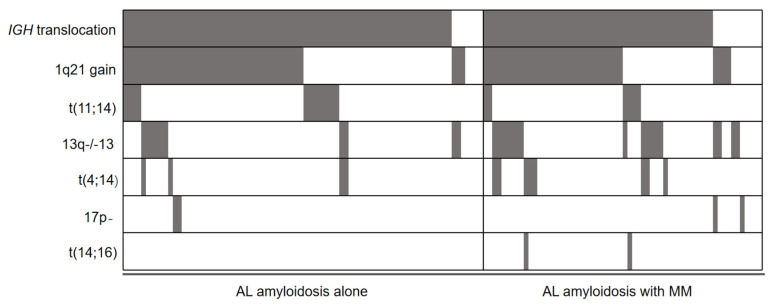
Distributions of FISH abnormalities. Gray shadow indicates positive results.

**Figure 2 jcm-12-01624-f002:**
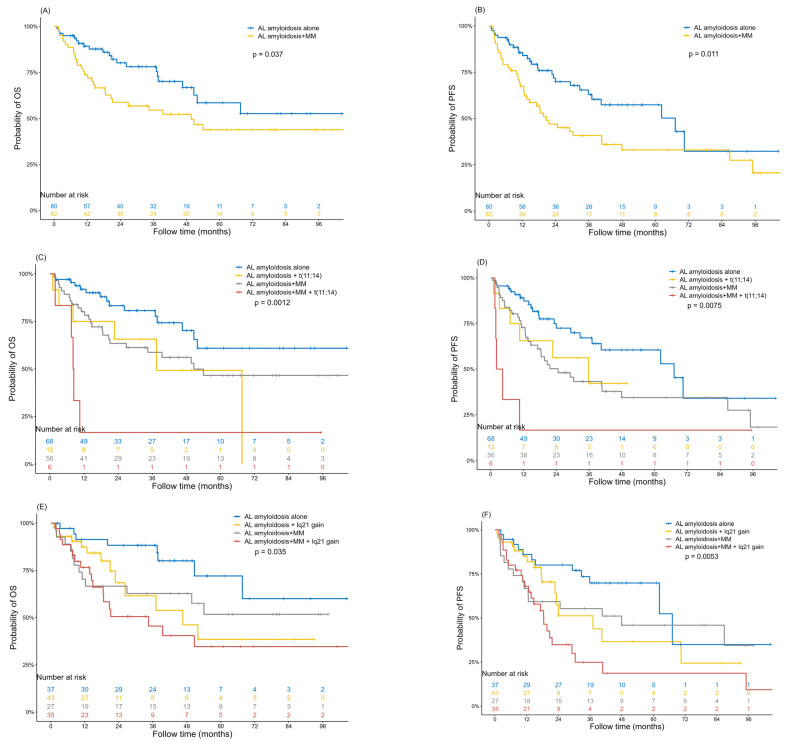
Kaplan–Meier curves. Overall survival (**A**) and progression-free survival (**B**) of AL amyloidosis patients with or without concurrent MM. Overall survival (**C**) and progression-free survival (**D**) of AL amyloidosis patients with or without concurrent MM, according to the presence or absence of t(11;14). Overall survival (**E**) and progression-free survival (**F**) of AL amyloidosis patients with or without concurrent MM, according to the presence or absence of 1q21 gain. Overall survival (**G**) and progression free survival (**H**) of AL amyloidosis patients with or without concurrent MM according to the presence or absence of 13q-; Overall survival (**I**) and progression free survival (**J**) of AL amyloidosis patients with or without concurrent MM according to the presence or absence of t(4;14). Abbreviations: AL, amyloid light chain; MM, multiple myeloma. Abbreviations: AL, amyloid light chain; MM, multiple myeloma. Abbreviations: AL, amyloid light chain; MM, multiple myeloma;13q-, 13q deletion.

**Table 1 jcm-12-01624-t001:** Baseline characteristics of AL amyloidosis patients, with or without concurrent multiple myeloma.

	AL Amyloidosis Alone (*n* = 80)	AL Amyloidosis with MM (*n* = 62)	*p* Value
Age (median, range), years	62.0 (53.0–67.3)	62.0 (56.2–67.8)	0.758
Gender, male, *n*(%)	94 (66.2)	56 (70.0)	0.363
M protein type, *n*(%)			0.010
IgG	23 (28.8)	28 (45.2)	
IgA	13 (16.3)	11 (17.7)	
IgD	0 (0.0)	4 (6.5)	
Light-chain only	40 (50.0)	18 (29.0)	
Others	4 (5.0)	1 (1.6)	
Lambda light-chain type, *n* (%)	63 (78.8)	46 (74.2)	0.662
Mayo stage 2004, *n* (%)			0.765
I	28 (43.1)	20 (38.5)	
II	22 (33.9)	17 (32.7)	
III	15 (23.1)	15 (28.9)	
Mayo stage 2012 IV, *n* (%)			0.007
I	35 (46.1)	9 (18.0)	
II	16 (21.1)	21 (42.0)	
III	14 (18.4)	13 (26.0)	
IV	11 (14.5)	7 (14.0)	
Hemoglobin (g/dL), median (IQR)	127.5 (116.2–141.3)	103.0 (85.3–121.8)	<0.001
Cr, median (IQR)	76.0 (61.0–112.0)	82.0 (63.0–149.0)	0.135
Albumin (g/dL), median (IQR)	28.0 (21.0–34.7)	29.3 (26.0–33.0)	0.431
β_2_ microglobulin (mg/dL), median (IQR)	2.63 (2.2–4.2)	4.86 (2.7–8.7)	0.001
Alkaline phosphatase (U/L), median (IQR)	86.0 (67.0–106.3)	73.0 (56.0–110.0)	0.459
LDH (U/L), median (IQR)	246.0 (194.5–345.8)	236.5 (170.8–332.8)	
NT-proBNP (pg/mL), median (IQR)	689.0 (139.8–2565.0)	1440.0 (412.0–5000.0)	0.033
dFLC (mg/dL), median (IQR)	122.9 (45.8–309.3)	487.9 (131.4–1496.8)	<0.001
Involved organ, *n* (%)			
Heart	36 (45.0)	22 (35.5)	0.331
Kidney	71 (88.8)	46 (74.2)	0.042
Liver	9 (11.3)	8 (12.9)	0.968

**Table 2 jcm-12-01624-t002:** The rates of FISH abnormalities.

	AL Amyloidosis Alone (n = 80)	AL Amyloidosis with MM (n = 62)	*p* Value
IGH translocation	73 (91.3)	51 (82.3)	0.175
t(11;14), n (%)	12 (15.0)	6 (9.7)	0.489
t(4;14), n (%)	4 (5.0)	8 (12.9)	0.169
t(14;16), n (%)	0 (0.0)	2 (3.2)	0.189
17p-, n (%)	2 (2.5)	2 (3.2)	1.000
13q-/-13, n (%)	10 (12.5)	17 (27.4)	0.042
1q21 gain, n (%)	43 (53.8)	35 (56.5)	0.880

## Data Availability

All data needed to evaluate the conclusions in the paper are present. in the paper. Additional data related to this paper may be requested from the authors.

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
