# Peer review of "The Landscape of Cytogenetic Aberrations in Light-Chain Amyloidosis with or without Coexistent Multiple Myeloma"

_jcm, 2023, doi:10.3390/jcm12041624_

Round 1

Reviewer 1 Report

This paper describes the prevalence of different cytogenetics of AL amyloidosis vs. AL amyloidosis plus myeloma.

The authors noticed that MM is associated with lower Hgb and shorter PFS and OS, as expected. However, in this study, t(11;14) in the MM+AL group was associated with a poor prognosis. 

Minor comments:

- Please explain why only 3 patients with AL (out of 80) and 6 with MM (out of 62) received ASCT. Was there a selection bias? Or is this expected in your center due to the availability of ASCT

-  Please spell out LSI. "LSI" was mentioned without explanation.

- I assume that none of the patients received venetoclax. If that is the case, please add it to the paper.

Reviewer 2 Report

I find the aim of the study relevant and interesting, and well described. Overall, i find the number of patients relatively small, especially when dividing into subcategories (some with 6 and 8 patients).

Comment and suggestions for the manuscript:

p.1, l.36-38: rephrase the sentence, you write "above patients" twice.

p.1, l.40: rememenber to explaine abbreviations (cIg-FISH)

p.1, l.45: the small minussign after 17p is nearly invisible, consider writing deletion og monosomi instead.

In the introduction section i lack the discussion on prognostic value of finding monosomi 13q with iFISH and not by conventional cytogenetics in patients with myeloma.

p.3, Table 1: results of male gender is missing in the table, age do not face the right line. I will like all p values!

I will like a comment on the difference in dFLC in the two groups. - in the discussion. I like, that you discuss on the surprisingly low percentage of t(11;14).

I find the median follow up surprisingly short.

p.4, l.141-143: I think you mixed up on the OS and PFS in the 2 groups.

p.6, l.195: T(11;14) - consider translocation(11;14) instead.

p.7, 1.section: do you think there is a difference i the t(11;14) mutation - current research show response to venetoclax in both amyloidosis and myeloma.

p.7, l. 210-211: melphalan-treated used twice - rephrase.

p.7, l. 215 "and" shoul be "a".

p.7, l. 216-217: "A potential... - the sentence shoul be rephrased. 
